# 3D-Printed EVA Devices for Antiviral Delivery and Herpes Virus Control in Genital Infection

**DOI:** 10.3390/v14112501

**Published:** 2022-11-11

**Authors:** Victor de Carvalho Rodrigues, Iara Zanella Guterres, Beatriz Pereira Savi, Izabella Thaís Silva, Gislaine Fongaro, Gean Vitor Salmoria

**Affiliations:** 1Nimma, Department of Mechanical Engineering, Federal University of Santa Catarina, Florianópolis 88040-900, SC, Brazil; 2Laboratory of Applied Virology, Department of Microbiology, Imunology and Parasitology, Federal University of Santa Catarina, Florianópolis 88040-900, SC, Brazil; 3Laboratory of Pharmacognosy, Department of Pharmaceutical Sciences, Federal University of Santa Catarina, Florianópolis 88040-900, SC, Brazil; 4Biomechanics Engineering Laboratory, University Hospital (HU), Federal University of Santa Catarina, Florianópolis 88040-900, SC, Brazil

**Keywords:** 3D printing, acyclovir, EVA, genital herpes

## Abstract

Herpes viruses are widespread in the human population and can cause many different diseases. Genital herpes is common and can increase the risk of HIV infection and neonatal herpes. Acyclovir is the most used drug for herpes treatment; however, it presents some disadvantages due to its poor oral bioavailability. In this study, some ethylene vinyl acetate devices with different acyclovir amounts (0, 10, and 20 wt.%) were manufactured by fused filament fabrication in two different geometries, an intrauterine device, and an intravaginal ring. Thermal analyses suggested that the crystallinity of EVA decreased up to 8% for the sample loaded with 20 wt.% of acyclovir. DSC, SEM, and FTIR analyses confirmed that the drug was successfully incorporated into the EVA matrix. Moreover, the drug release tests suggested a burst release during the first 24 h followed by a slower release rate sustained up to 80 days. Biological assays showed the biocompatibility of the EVA/ACV device, as well as a 99% reduction in vitro replication of HSV-1. Finally, the EVA presented a suitable performance for 3D printing manufacturing that can contribute to developing personalized solutions for long-term herpes treatment.

## 1. Introduction

Herpes simplex virus (HSV) is a DNA virus widespread in the human population due to its effective transmission. Among the different herpesvirus types, HSV-1 and HSV-2 are responsible for many important diseases such as orolabial herpes, encephalitis, and genital herpes [1,2,3]. After primary infection, latency occurs in specific neurons and the latent virus can be periodically reactivated [2,4,5]. The mechanisms responsible for this reactivation are not completely understood but are mainly related to hyperthermic stress, UV irradiation exposure, and hormone treatment [2,5,6].

Acyclovir (ACV) is a highly efficient antiviral agent against HSV. It is a guanosine analog inhibiting the viral DNA polymerase and blocking viral replication [1,7]. The conventional genital herpes treatment is based on oral or topical administration of antivirals [8]. However, even with few significant adverse effects, ACV presents a low oral bioavailability and a limited absorption into the gastrointestinal tract [7,9]. The acyclovir prodrug valacyclovir has improved this limited bioavailability, reducing dose frequency [10,11]. However, for suppressive therapy, daily administration is still required, and a lack of patient compliance can reduce treatment efficacy [10]. Therefore, to reduce these drawbacks, many alternatives have been explored, especially some drug delivery systems (DDS) such as nanoparticles [9,12], electrospinning drug-loaded fibers [13,14], hydrogels [15], and intravaginal rings (IVR) [16,17].

Additive manufacturing techniques have been improved during the last few years [18,19]. Fused filament fabrication (FFF) is one of the most popular methods especially given the diversity of thermoplastics available to use [20,21]. The three-dimensional geometry file is produced using a computer-aided design (CAD) software and the printer produces the samples by heating the filaments and depositing successive layers until the final product is completed [19,20]. Nowadays, the use of 3D printing is growing fast in biomedical and pharmaceutical fields [21,22,23,24]. This process allows the production of personalized medication or DDS according to patient needs or disease complexity [22,25]. However, large-scale production and the lack of regulatory guidelines are still some of the main challenges for this disruptive technology [26].

The EVA is a non-degradable copolymer of ethylene and vinyl acetate (VA). The VA values generally range from 0 to 50% and affect some important properties such as transition temperatures, crystallinity, hardness, and polarity [27]. It is widely used in transdermal systems with functional applications due to its biocompatibility and formulation variety [27,28,29]. For DDS manufacturing, the hot-melt extrusion (HME) process is the most used, especially due to the EVA low glass transition temperature that allows great compatibility with the most diverse active pharmaceutical ingredients (API) [29,30]. Among the EVA-based products, Nuvaring^®^ is an intravaginal ring available on the market manufactured by HME for contraceptive applications [31,32]. Some other EVA drug delivery systems are reported in the literature including intravaginal rings [16,33], oral sustained release tablets [34], and 3D-printed devices [28,35].

In this study, EVA drug-loaded filaments were fabricated by hot-melt extrusion with different acyclovir amounts (0, 10, and 20 wt.%). EVA/ACV devices were manufactured by fused filament fabrication (FFF) using temperatures below the ACV melting point. Two geometries were printed, an intrauterine device and an intravaginal ring. The morphological, physicochemical properties and drug release mechanisms were evaluated. Moreover, cytotoxicity assay and reduction replication in vitro of HSV-1 were performed. Finally, the feasibility of these devices for the long-term treatment of genital herpes is discussed.

## 2. Materials and Methods

### 2.1. Materials

Ethylene-vinyl acetate (EVA) copolymer pellets (ATEVA^®^ 1615) were supplied by Celanese (USA) with 16% VA, density of 0.937 g/cm^3^, melt index of 15 g/10 min (190 °C/2.16 kg) and melting temperature of 90 °C. Acyclovir was purchased from Zhejiang Charioteer Pharmaceutical Company (China) in the form of powder and molar mass of 225.21 g/mol.

### 2.2. Hot-Melting Extrusion (HME) Process

EVA pellets and ACV powder were dried in a vacuum oven (MA030, Marconi, Brazil) at 60 °C for two hours to reduce moisture content. Filaments were prepared with different acyclovir weight percentages (0, 10, and 20 wt.%) using a single-screw extruder (Filmaq3D CV, Filmaq, Brazil) with an extrusion die of 1.75 mm. Processing parameters were the same for all the samples and configured with an extrusion temperature of 140 °C and screw speed of 10 rpm. The extruded filaments were cooled in ambient conditions and the diameter was controlled at 1.65 ± 0.10 mm to ensure suitable printing.

### 2.3. 3D printing by Fused Filament Fabrication (FFF)

Samples were printed by the fused filament fabrication method using the S3 (Sethi 3D, Brazil) printer in two different shapes, an IUD (46 × 33 × 2 mm) and an intravaginal ring (54 × 4 mm), as shown in Figure 1.

Several preliminary tests were carried out to optimize the printing parameters. For all the samples the printing speed, nozzle diameter, layer height, and line width were the same and configured to 2.5 mm/s, 0.4 mm, 0.25 mm, and 0.45 mm, respectively. The build platform temperature was set at 25 °C and it was covered with a PEAD film to increase the adhesion between the EVA and the build plate. The printing temperature was configured depending on the composition and geometry as described in Table 1.

### 2.4. Infrared Spectroscopy

Fourier-transform infrared (FTIR) spectroscopy was performed using a Frontier MIR/NIR (PerkinElmer, Waltham, MA USA) spectrophotometer in attenuated total reflectance (ATR) mode scanning from 4000 cm^−1^ to 400 cm^−1^. For each analysis, 32 scans were acquired at a resolution of 2 cm^−1^ to characterize the polymer and drug absorbance peaks and verify the incorporation of acyclovir.

### 2.5. Scanning Electron Microscopy

Briefly, 3D-printed EVA samples of each formulation were examined by scanning electron microscopy (SEM) using a JSM-6390LV (JEOL, Tokyo Japan) microscope to analyze the surface and cross-section morphology. Energy dispersive spectroscopy (EDS) was also performed to identify drug particles.

### 2.6. Differential Scanning Calorimetry

Thermal analyses were performed using a differential scanning calorimeter DSC 6000 (PerkinElmer, USA) from −40 to 280 °C followed by cooling from 280 to −40 °C with a heating and cooling rate of 10 °C/min and a nitrogen flow of 20 mL/min. The average sample weight was 5 mg and the tests were carried out in duplicate. The degree of crystallinity (χ_c_) of 3D-printed EVA samples were calculated according to Equation (1):(1)$$\chi_{c}=\frac{\Delta H_{m}}{\Delta H_{m0}}\times 100$$
where Δ*H_m_* is the melting enthalpy obtained from DSC analyses and Δ*H_m_*_0_ is the melting enthalpy of a pure crystal of linear polyethylene (293 J/g) [36].

Acyclovir real weight percentage (%*ACV*) was calculated based on Equation (2), where Δ*H_ms_* is the melting enthalpy of drug-related peak for the EVA/ACV samples and Δ*H_ms_*_0_ is the melting enthalpy measured for the pure ACV samples (117 J/g).
(2)$$\%ACV=\frac{\Delta H_{ms}}{\Delta H_{ms0}}\times 100$$

### 2.7. Drug Release Tests

To evaluate the drug release behavior, 3D-printed specimens with known dimensions and drug concentration were placed in sealed test tubes with 15 mL of phosphate buffer solution (PBS, pH = 7.4). The tubes were shaken horizontally in a Dubnoff bath (Quimis SA, Santa Catarina Brazil) at 60 rpm and 37 ± 0.5 °C to minimize the boundary effects. PBS solution was periodically removed and replaced by a fresh solution. After suitable dissolution, the solution was analyzed using a UV–Vis spectrophotometer UV-5200 (Global Trade Technology, São Paulo Brazil) at λ_max_ of 251 nm. The calibration curve was previously constructed for ACV on PBS for different concentrations ranging from 0.001 to 0.05 mg/mL, with a linear behavior in this region (R^2^ = 0.9989). These tests were performed in triplicate.

### 2.8. Cytotoxicity and Antiviral Assay

The samples were submitted for UV disinfection. The disinfection process was confirmed by inoculation in Luria Broth (specific for bacterial growth) and Sabouraud agar (specific for fungal growth). The cytotoxic screening was conducted using Vero cells (ATCC CCL-81-VHG), an immortalized culture obtained from fibroblasts derived from African green monkey kidneys. The cell culture was grown in Minimal Essential Medium (MEM) supplemented with 10% fetal bovine serum (FBS; Gibco, Grand Island, NY) and maintained at 37 °C and 5% CO_2_ in a humidified atmosphere. The cytotoxicity of the EVA/ACV devices was evaluated by the colorimetric assay of sulforhodamine B (SRB) [37]. Briefly, Vero cells were seeded (3.33 × 10^5^ cells/mL in 24-well plates) and after 24 h of incubation, the devices were added in each well. Some cavities of each plate did not receive any sample and were used as cell control. After 48 h of incubation at 37 °C and 5% CO_2_, 10% of trichloroacetic acid (TCA) was added to each well in order to fix the cells. After 1 h, the TCA solution was removed and SRB was added. After 30 min, the plates were washed with a watery solution of 1% acetic acid to remove unbound SRB and allow visualization of viable cells.

In order to determine the potential antiviral effect of the devices, Vero cells were seeded in 24-well plates at a concentration of 3.33 × 10^5^ cells/mL. After the formation of a confluent cell monolayer, cells were infected with HSV-1 quantified at 5 Log plaque forming units/mL (PFU/mL) and devices were added to each well (pure EVA and EVA/ACV devices). Plates of virus-infected cells containing the devices in the presence and absence of ACV were incubated at 37 °C and 5% CO_2_. After 48 h, the devices were removed and the cell supernatants were collected, containing the virus particles from the replication process. Then, the supernatants were titrated according to Burleson et al. (1992) using plaque assay methods. For this, 24-well plates with Vero cells (3.33 × 10^5^ cells/mL) were inoculated with the replication supernatants subjected to 10-fold serial dilutions and inoculated into the cells. The plates were incubated for 48 h and the cells were fixed and stained with naphthol blue-black for quantitation of viral plaques using a stereomicroscope. All experiments were performed in independent triplicates.

## 3. Results and Discussion

### 3.1. Morphological Characterization

Pure EVA 3D-printed devices were translucent while the EVA/ACV were white and opaque. The same behavior was observed for both the IUD and the IVR. This color suggests that the ACV was not dissolved or melted on the matrix, considering the scanning SEM studies [38], which was expected as the extrusion and printing temperatures were lower than ACV melting temperature.

The SEM micrographs of EVA and EVA/ACV intravaginal rings are shown in Figure 2 (IUD’s micrographs are shown in Appendix A). The surface roughness increases with the ACV content for both IVR and IUD. It can be noticed that the cross-section showed a greater concentration of drug particles compared to the surface. The presence of ACV particles in the EVA-P20 samples was preeminent, confirming the difference in terms of drug content. Moreover, the ACV particle aggregates (indicated by red arrows) increase porosity in the EVA matrix.

The EDS was performed on different cross-sectional regions of EVA/ACV samples, as shown in Figure 3, to determine the chemical composition of these devices. ACV has nitrogen atoms in its chemical structure, while EVA is composed only of carbon, hydrogen, and oxygen. The identification of ACV particles was possible by comparing the nitrogen content.

The EDS analyses found higher concentrations of nitrogen for drug particles (17.62 ± 3.15 wt.% at points 1 and 2) than in the EVA matrix (5.3 ± 0.13 wt.% at points 3 and 4), confirming the presence of ACV dispersed throughout the EVA matrix. Although there is no nitrogen in the composition of EVA, the small percentage of this element reported in the EVA matrix was due to the drug particles close to the analyzed region. Furthermore, in EDS analysis the carbon and nitrogen peaks overlap (as shown in Figure 3B), which reduces the accuracy of nitrogen quantification [39].

### 3.2. Thermal Analysis

The DSC analyzes were performed to characterize the thermal behavior of acyclovir, EVA, and EVA/ACV devices. Thermal analysis was carried out from −40 to 280 °C to investigate the drug incorporation. The thermograms of the heating cycle are presented in Figure 4. EVA has a complex crystalline structure with different morphologies [36,40]. Therefore, the melting region presents two peaks. The first (T_m1_) is related to the region with a lower degree of organization, whereas the second (T_m2_) refers to the pure polyethylene structure [36]. Based on the results and following Equation (1), the melting enthalpy (Δ*H_m_*) reduced with the ACV addition as well as the crystallinity (χ_c_) degree decreased up to 8% for the EVA-P20.

Drug incorporation into 3D-printed samples was also confirmed by DSC analysis. Samples with 10 wt.% (EVA-P10) and 20 wt.% (EVA-P20) of ACV presented an endothermic event at the same temperature of acyclovir melting point (257.8 ± 0.7 °C), as shown in Figure 4. Furthermore, this peak was higher for the EVA-P20 sample than the EVA-P10, showing the different drug concentrations between them. Following Equation (2) and based on the melting enthalpy (Δ*H_macv_*) of pure ACV and the drug-related melting peaks of EVA-P10 and EVA-P20 samples, it was possible to estimate the real drug weight percentage on each device. According to the results described in Table 2, the real ACV amount is approximately 25% lower than the expected. This reduction is due to the losses during the HME process. As the extrusion temperature is below the ACV melting point (~400 °C), a portion of drug powder remains retained on the screw and equipment walls. Hollander and collaborators reported a similar problem using hot-melt extrusion [41].

Crystallization temperature is an important parameter for the fused filament fabrication process. Slower cooling rates and higher printing temperatures tend to keep the printed layers above glass transition temperature for longer periods, which improves the adhesion between them. Several tests were performed to find the best printing setup for each sample. It was noticed that the ACV addition reduces the adhesion of the layers. For this reason, the printing temperature was higher according to the ACV content. The results obtained in the cooling cycle are described in Table 3. For the pure EVA samples, the crystallization enthalpy (Δ*H_c_*) is higher for EVA-pellet than pure EVA 3D-printed sample (EVA-P). The same behavior was observed for both melting enthalpy (Δ*H_m_*) and polymer crystallization (χ_c_). It suggests that, although the ACV particles may introduce defects into the EVA matrix and reduce its crystallinity, the HME process is also responsible for the changes in the EVA crystalline structures.

### 3.3. FTIR Analysis

The successful loading of ACV in EVA devices was verified through the identification of characteristic absorbance peaks of the drug by ATR-FTIR analysis. ACV shows an absorbance region between 3200 cm^−1^ and 3600 cm^−1^ mainly due to N-H primary and secondary amine stretching or to O-H stretching [14,42]. The 1738 cm^−1^ and 1634 cm^−1^ peaks are related to the C=O stretching and N-H primary amine bending, respectively [14,43]. Moreover, the 1542 cm^−1^ band represents C=N and C-N stretching [42]. The absorbance bands at 1186 cm^−1^ and 1107 cm^−1^ are associated with the bending of amine groups and C-O stretching [42]. The 785 cm^−1^ peak occurs due to the out-of-plane bending of C-H groups and 685 cm^−1^ to the N-H wagging [12,14]. The EVA characteristic FTIR spectra show some absorbance peaks between 2928 cm^−1^ and 2849 cm^−1^ related to the asymmetric and symmetric stretching of -CH2- groups, respectively [44]. The 1736 cm^−1^ peak is related to the C=O vibrations [28,44]. Furthermore, the bands at 1463 cm^−1^ and 1370 cm^−1^ are associated with deformation in the plane of -CH2- and -CH3- in the plane [45]. Finally, the peak at 718 cm^−1^ refers to long chains -CH2- present in polyethylene (PE) [45].

For the samples loaded with ACV, some drug characteristic bands that were absent in the unload EVA devices spectrum were identified, as described in Figure 5. For instance, the signals observed between the 3440 cm^−1^ and 3190 cm^−1^ for the EVA-P10 and EVA-P20 are related to ACV amine stretching. Moreover, the peaks located at 1634 cm^−1^ and 1542 cm^−1^ also confirmed the presence of acyclovir. Some other characteristic bands of ACV may be observed between 1186 cm^−1^ and 685 cm^−1^ for EVA/ACV samples. Furthermore, the signal of these peaks was greater for the EVA-P20 sample, confirming not just the ACV incorporation but also the different drug content between EVA-P10 and EVA-P20.

### 3.4. Drug Release Tests

The drug release profiles of EVA/ACV samples are described in Figure 6. The results showed a burst release on the first day followed by a slower release rate sustained up to 80 days. The sample with 10 wt.% of ACV (EVA-P10) released about 821.90 ± 168.118 μg/g while the device with 20 wt.% of acyclovir released 1306.28 ± 280.42 μg/g during the first 24 h. This accelerated initial release is characteristic of EVA devices and is also reported by other authors [28,34,46]. It occurs due to the drug particles available on the surface of the device in contact with the release medium. For genital herpes treatment, this behavior can be helpful as the faster initial release can control symptomatic episodes while the sustained release prevents further events. Moreover, some studies have reported that the first few hours of viral expansion are crucial for HSV treatment [47]. Therefore, the efficiency of constant prophylactic ACV administration is greater than acute treatment, mainly because the antiviral is present from the beginning of viral expansion.

Mathematical modeling is widely used to evaluate the drug release profile in the pharmaceutical industry. It is an essential tool to improve the research and development processes, predicting release behavior, and reducing costs and number of tests performed [48]. The drug release mechanisms of EVA/ACV devices were evaluated considering the mathematical models described in Table 4. Both samples presented an accurate R^2^ for the Korsmeyer–Peppas and Weibull method. The first is used to characterize the release mechanisms based on the *n* (diffusion exponent) parameter. Both samples presented a *n* < 0.45 that corresponds to a Fickian diffusion mechanism [49]. On the other hand, Weibull’s method can be used to compare the release profile of matrix-based devices, as the ones used in this work. The *β* < 0.35 indicated that the cumulative release curve has a parabolic behavior with a steeper initial slope, which helps to explain the initial burst release [50,51,52].

### 3.5. Cytotoxicity and Antiviral Assay

Biocompatibility is an essential requirement for drug-delivery devices. For this reason, the cytotoxicity of EVA/ACV printed samples was investigated. EVA devices generally present good biocompatibility and are widely used in biomedical and pharmaceutical applications [28,29,53]. Compared with cell control, the cell viability with treated devices was 100%, without any change in cell morphology. Therefore, this result demonstrated that the printed devices with acyclovir content did not show cytotoxicity, confirming its biocompatibility.

In the antiviral activity assay performed in independent triplicate, the results demonstrated that the treated devices were efficient against HSV-1 infection. The reduction in the cytopathic effect was observed since the first viral concentration that was tested. The reduction in the viral load from 5 Log10 UFP/mL to 3 Log10 UFP/mL, indicates 2 Log10 reductions or a 99% reduction in the viral load (Figure 7).

## 4. Conclusions

This work presented a new alternative to preparing EVA/ACV devices via fused filament fabrication for sustained drug release of antiviral drugs. SEM images revealed that the surface roughness increases with the ACV content. Furthermore, the drug particles are mostly located inside the matrix. EDS, FTIR, and DSC analysis confirmed the successful loading of acyclovir in EVA devices. DSC also showed that the crystallinity decreases with the ACV addition. Moreover, comparing the crystallinity of EVA pellets and the pure EVA 3D-printed devices (EVA-P), the HME process seems to influence EVA structures and reduce their crystallinity. The drug release profile showed a burst release during the first 24 h followed by a slower rate of sustained release for up to 80 days. This behavior can be helpful as the faster initial release can control symptomatic episodes while the sustained release prevents further events. The biological assays demonstrated the potential biocompatibility of the printed devices, being also able to reduce by 99% the in vitro replication of HSV-1. The results obtained in this work demonstrated the potential of these printed devices for long-term genital herpes treatment. Therefore, it indicates the feasibility of the fused filament fabrication process to produce personalized and more efficient solutions for the biomedical and pharmaceutical industries. For future research, new materials can be tested to compare performance between different devices and clinical trials should be considered to validate the experiments performed.

## Figures and Tables

**Figure 1 viruses-14-02501-f001:**
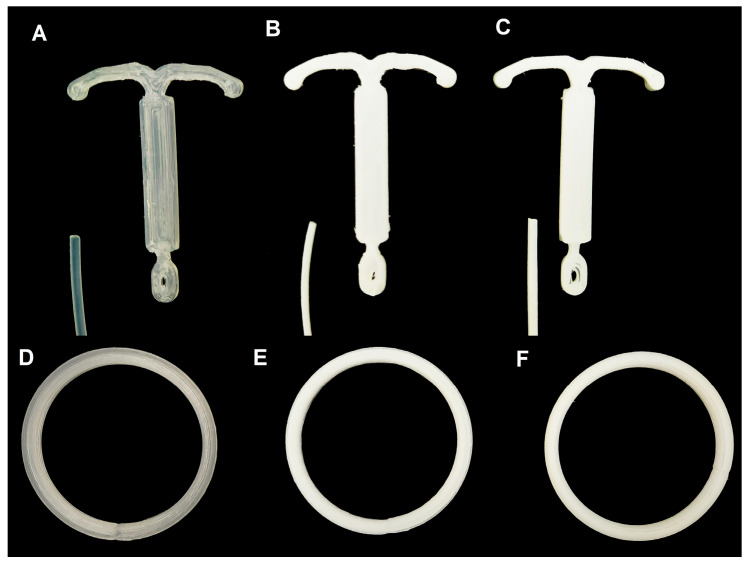
IUDs (**top**) and intravaginal rings (**bottom**) manufactured by fused filament fabrication using pure EVA (**A**,**D**) and EVA/ACV with 10 wt.% (**B**,**E**) and 20 wt.% (**C**,**F**) of acyclovir.

**Figure 2 viruses-14-02501-f002:**
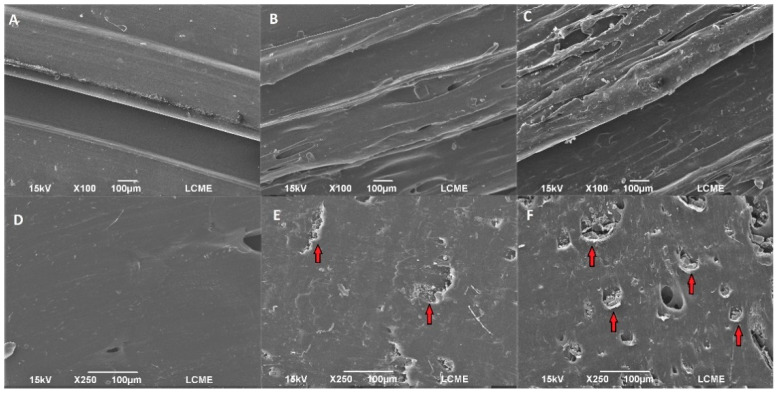
SEM images of surface (**top**) and cross-section (**bottom**) of 3D-printed intravaginal rings of EVA-P (**A**,**D**), EVA-P10 (**B**,**E**), and EVA-P20 (**C**,**F**).

**Figure 3 viruses-14-02501-f003:**
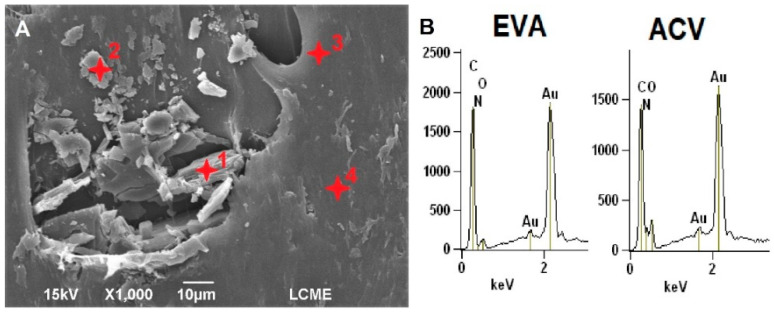
SEM images of the cross-section EVA-P20 (**A**) intravaginal rings samples with the EDS regions and the respective EDS spectra for carbon, oxygen, and nitrogen content in the EVA matrix and ACV particles (**B**).

**Figure 4 viruses-14-02501-f004:**
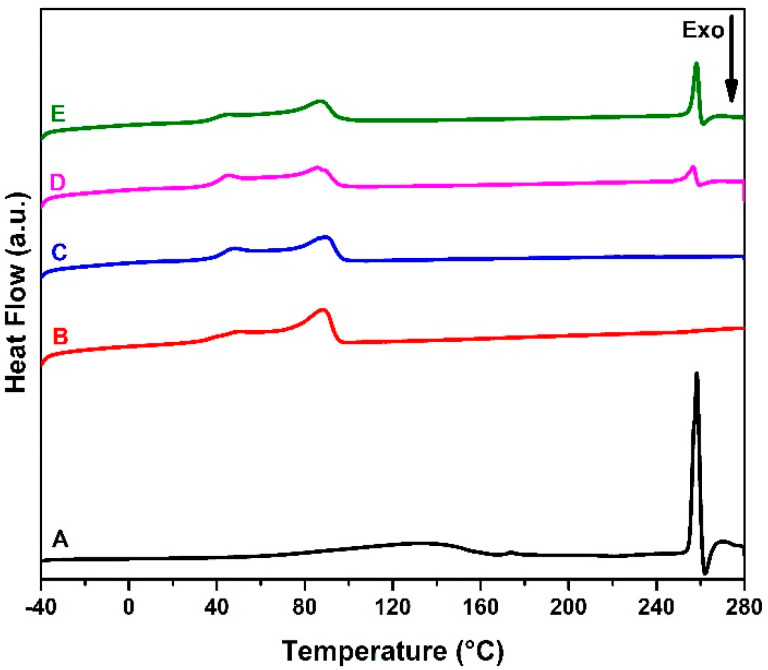
DSC thermograms of ACV (A), EVA pellet (B) and 3D-printed samples EVA-P (C), EVA-P10 (D), and EVA-P20 (E).

**Figure 5 viruses-14-02501-f005:**
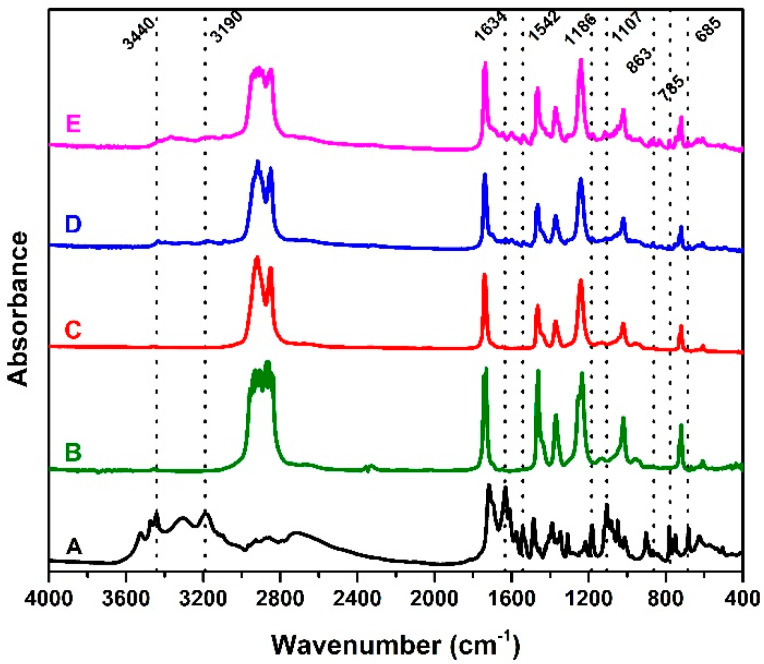
ATR-FTIR spectra of ACV (A), EVA pellet (B) and 3D-printed samples EVA-P (C), EVA-P10 (D), and EVA-P20 (E).

**Figure 6 viruses-14-02501-f006:**
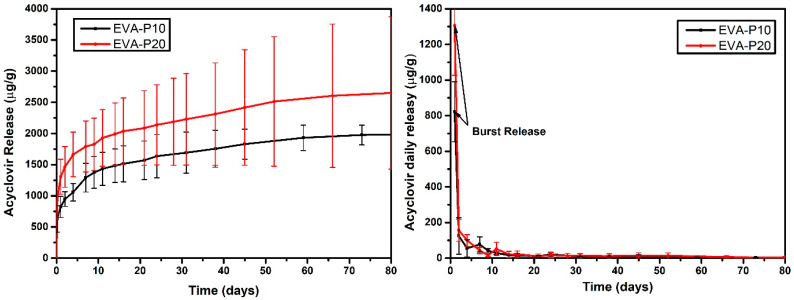
Acyclovir daily (**right**) and accumulated (**left**) release as function of time for EVA-P10 and EVA-P20 samples.

**Figure 7 viruses-14-02501-f007:**
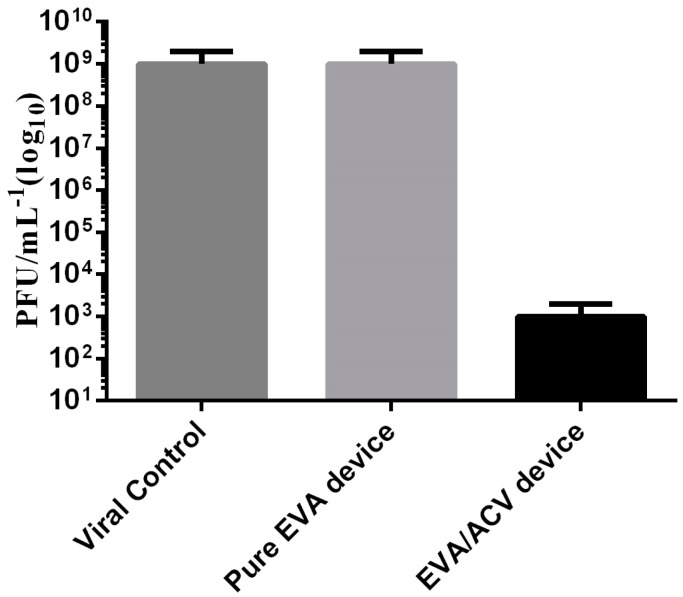
Antiviral activity against HSV-1, comparing the viral control, pure EVA device, and EVA device treated with ACV (*n* = 3).

**Table 1 viruses-14-02501-t001:** Printing temperatures of EVA and EVA/ACV devices.

Sample	% ACV (wt.%)	Printing Temperature (°C)
EVA-P	0	160
EVA-P10	10	170
EVA-P20	20	190

**Table 2 viruses-14-02501-t002:** Thermal behavior data obtained from DSC heating cycle for EVA and EVA/ACV samples.

Sample	T_m1_ (°C)	T_m2_ (°C)	Δ*H_m_* (J/g)	χ_c_ (%)	T_macv_ (°C)	Δ*H_macv_* (J/g)	%ACV (wt.%)
EVA-Pellet	49.9 ± 0.2	88.9 ± 0.8	89.1 ± 4.4	30.4 ± 1.5	-	-	-
EVA-P	48.4 ± 0.3	89.2 ± 0.3	77.9 ± 2.5	26.6 ± 0.9	-	-	-
EVA-P10	45.4 ± 0.1	86.7 ± 0.9	70.7 ± 0.7	24.1 ± 0.3	256.7 ± 0.1	9.0 ± 1.0	7.7 ± 0.8
EVA-P20	45.2 ± 0.3	86.4 ± 0.4	66.4 ± 7.3	22.7 ± 2.5	258.2 ± 0.1	17.3 ± 1.1	14.8 ± 0.9

**Table 3 viruses-14-02501-t003:** Thermal behavior data obtained from DSC cooling cycle for EVA and EVA/ACV samples.

Sample	T_c1_ (°C)	T_c2_ (°C)	Δ*H_m_* (J/g)
EVA-Pellet	43.4 ± 0.8	75.0 ± 0.8	−72.4 ± 10.0
EVA-P	51.8 ± 0.8	75.8 ± 0.1	−50.7 ± 1.3
EVA-P10	53.8 ± 0.1	75.3 ± 0.1	−40.3 ± 1.0
EVA-P20	44.3 ± 0.1	75.6 ± 0.1	−49.3 ± 9.1

**Table 4 viruses-14-02501-t004:** Drug release parameters of EVA/ACV samples fitted by mathematical kinetic models.

Material	Zero-Order *	Korsmeyer–Peppas		Weibull		
	R^2^	R^2^	*n*	R^2^	A	β
EVA-P10	89.52	98.49	0.17	97.69	113.34	0.17
EVA-P20	87.98	99.72	0.16	99.79	156.47	0.17

* R^2^ disregarding the first day (after the “burst release”).

## Data Availability

Not applicable.

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
