# Peer review of "3D-Printed EVA Devices for Antiviral Delivery and Herpes Virus Control in Genital Infection"

_viruses, 2022, doi:10.3390/v14112501_

Round 1

Reviewer 1 Report

This group are detailing a study in which they generate 3 D printing to make Ethylene vinyl acetate intravaginal devices which are impregnated with the antiviral ACV at 10 and 20%.   They test the devices in a number of ways for their physical make up, and show that they contain the antiviral in an active form.   They propose that they may have use on prolonged treatment for intravaginal use.   Overall, this seems a rather preliminary study. What they do NOT do is show the antiviral effects in any kind of model and compare it to simple topical treatment with ACV creams and currently available options. There are also several additional concerns that must be addressed, because as the paper stands, it leaves much to be desired to be a contribution of importance

1.       There are problems with the English grammar   The title is just the start ( it does not make sense) . the abstract has many and the entire text. This article needs the assistance of an English editor.

2.       2  there is a need for a stronger rationale for the study, particularly given the very rapid and not prolonged release of the ACV in the first 24 hrs.   the rationale for the study is to obtain prolonged release, but it does not seem like they have achieved it. , The argument that ACV is poorly bioavailable is quite weak, since valacyclovir is very orobioavailable.  . There is no acknowledgment or even mention of Val acyclovir, which has overcome virtually all of the prolonged use and Oro bioavailability problems. thre is no mention of the use of topical ACV.  ACV is also generally reasonably soluble compared to many other drugs, so highlighting its insolubility is a little biased and not an entirely correct statement. 

3.        Line 32 the statement that of the herpesviruses, HSV-1 and 2 are the most common is quite wrong.   VZV, CMV, EBV and HHV6 and 7 are all more common and widespread than either of the two simplex viruses than HSV-1 and HSV-2.

4.       36/37 if the virus is latent, it cannot be transmitted.  Transmission can only occur when there is active virus replication and production of infectious virus (which may, however, be asymptomatic/subclinical, which is what i think they mean to say)

5.       Line 42 ACV works primarily by inhibiting the viral DNA polymerase, though it also works by DNA chain termination

6.        Line 71 What is the ACV "melting point"? This makes little sense to me. More important is the drug inactivation temperature of ACV? I note that extrusion requires quite high temperatures in the 160-190oC range (table1). While they show that some ACV remains in the printed devices, there is no quantification of how much of the ACV remains active versus inactive. This is needed, much more so than the relatively irrelevant other physical assessments done.  In particular, figure 3B makes no sense and is not informative. 202 -208. I found this data to be very confusing and not at all clear or well explained. ACV is there, yes. But how much of it has survived the extrusion process?  Figure 5 is likewise not informative.  What do ATR-FTIR data show?

7.       I am concerned about the data in figure 6 it appears the vast majority of the ACV is immediately released on day 1. It does suggest tiny amounts are released later after day 1, but are these at levels of any effective use?    In addition, the release experiment is somewhat flawed as the study tried to evaluate what would be released in a “in use” setting.  But the study was done at pH 7.4.  the vaginal pH is generally considered to be a more acidic ranging from 3.5-5.   At minimum the study should be redone at this relevant pH as for amounts released over time (right figure 6 panel) maybe a log graph might be better, as nothing can be seen to be released after day 20. How does this compare with a dose of topical ACV or ACV in a liquid matrix, such as a cream?  There are no studies done to indicate an improvement over current protocols and uses.  and the lack of any animal model testing demotes the study importance

Author Response

Response to Reviewer 1 Comments:

1. There are problems with the English grammar   The title is just the start ( it does not make sense). The abstract has many and the entire text. This article needs the assistance of an English editor.

Answer: The English grammar was reviewed, and the title changed to “3D printed EVA devices for antiviral delivery and herpes virus control in genital infection”.

2. There is a need for a stronger rationale for the study, particularly given the very rapid and not prolonged release of the ACV in the first 24 hrs.   the rationale for the study is to obtain prolonged release, but it does not seem like they have achieved it. , The argument that ACV is poorly bioavailable is quite weak, since valacyclovir is very orobioavailable.  . There is no acknowledgment or even mention of Val acyclovir, which has overcome virtually all of the prolonged use and Oro bioavailability problems. thre is no mention of the use of topical ACV.  ACV is also generally reasonably soluble compared to many other drugs, so highlighting its insolubility is a little biased and not an entirely correct statement. 

Answer: We appreciate the reviewer’s contribution. Some information about both topical and valacyclovir administration were included in the introduction section (L). However, the main disadvantage of oral or topical administration is the daily doses required, especially during suppressive therapy. Although the vaginal route may present better absorption than the gastrointestinal tract, the main advantage of using these drug delivery devices is to reduce some problems such as the lack of patient compliance during the treatment of recurrent cases as multiple daily doses are required for long periods. Moreover, the intravaginal and intrauterine devices may be more efficient to prevent recurrent cases as they are constantly releasing the antiviral into the patient’s body. This can also be relevant in cases of pregnancy, to prevent neonatal herpes. Furthermore, as mentioned in L 284-285 “[…] the first few hours of viral expansion are crucial for the HSV treatment. Therefore, the efficiency of constant prophylactic ACV administration is greater than acute treatment, mainly because the antiviral is present from the beginning of viral expansion”.

As the reviewer described, the valacyclovir and other drugs already improved the limited bioavailability of acyclovir, as well as many other solutions described in the introduction section L 47-50: “[…] different drug delivery systems (DDS) such as nanoparticles, electrospinning drug-loaded fibers, hydrogels and intravaginal rings (IVR)”. However, the main goal of this work was not just to propose a solution to the poor bioavailability of acyclovir, but to develop alternatives produced through additive manufacturing to genital herpes treatment. The use of fused filament fabrication has been growing as well as the interest of pharmaceutical and biomedical industry, especially as this process allows the production of personalized medication or DDS according to patient needs or disease complexity. However, there is just a few works focusing on drug release monitoring of 3D printed devices. The results were promising especially in terms of the EVA performance for drug release applications. Therefore, the authors believe that these results could contribute not just to the development of alternative treatments to genital herpes, but also to demonstrate the feasibility of the additive manufacturing to produce suitable devices for drug release applications.

We accepted the advice about acyclovir water solubility and replaced the sentence as follows L  42-44.

  1. Line 32 the statement that of the herpesviruses, HSV-1 and 2 are the most common is quite wrong.   VZV, CMV, EBV and HHV6 and 7 are all more common and widespread than either of the two simplex viruses than HSV-1 and HSV-2.

Answer: The suggestion was accepted. The information was corrected as follows L 32-34: “Among the different herpesvirus types, HSV-1 and HSV-2 are responsible for many important diseases such as orolabial herpes, encephalitis, and genital herpes […]”

  1. 36/37 if the virus is latent, it cannot be transmitted.  Transmission can only occur when there is active virus replication and production of infectious virus (which may, however, be asymptomatic/subclinical, which is what I think they mean to say)

Answer: The suggestion was accepted. The information was removed.

  1. Line 42 ACV works primarily by inhibiting the viral DNA polymerase, though it also works by DNA chain termination

Answer: The information was improved to L 38-39: “Acyclovir (ACV) is a highly efficient antiviral agent against HSV. It is a guanosine analog inhibiting the viral DNA polymerase and blocking viral replication”.

  1. Line 71 What is the ACV "melting point"? This makes little sense to me. More important is the drug inactivation temperature of ACV? I note that extrusion requires quite high temperatures in the 160-190 °C range (table1). While they show that some ACV remains in the printed devices, there is no quantification of how much of the ACV remains active versus inactive. This is needed, much more so than the relatively irrelevant other physical assessments done.  In particular, figure 3B makes no sense and is not informative. 202 -208. I found this data to be very confusing and not at all clear or well explained. ACV is there, yes. But how much of it has survived the extrusion process?  Figure 5 is likewise not informative.  What do ATR-FTIR data show?

Answer: We appreciate the reviewer’s questions. The acyclovir melting temperature was found to be 257.8 ± 0.7 °C (L 224), which is in accordance with what was reported. The melting point corresponds to the temperature where drug particles change from a solid to a liquid state. We believe that this information is relevant especially because the printing parameters, the drug release mechanisms/kinetics and many important properties of the manufactured devices are dependent on the drug’s physical state. For instance, drug delivery devices manufactured by hot-melt extrusion using temperatures above the API melting point present different release behavior than devices containing solid drug particles dispersed throughout the matrix, as studied in this work.

It is important to notice that the acyclovir generally starts its thermal degradation at temperatures above 400 °C, as described by Celebioglu & Uyar (2021) and Shamsipur et al. (2013) Therefore, there was no thermal degradation of ACV during the extrusion and/or printing process. Thermal analysis was carried from -40 to 280 °C to investigate the drug incorporation. The melting temperature is the region of interest to check this information and can be used to estimate the drug percentage of each device as described in L 129-131. Therefore, from the DSC data it was possible to quantify the acyclovir real percentage after the manufacturing process as described in Table 2 (% ACV). Is it worth mentioning that the ACV losses during the extrusion are not due to any thermal degradation but due to the particles retained into the equipment cavities (L 231-233). Furthermore, the SEM images showed that drug particles were in the solid state and were located mainly inside the sample which is a characteristic of matrix devices.

References:

Celebioglu, A., & Uyar, T. (2021). Electrospun formulation of acyclovir/cyclodextrin nanofibers for fast-dissolving antiviral drug delivery. Materials Science and Engineering C, 118. https://doi.org/10.1016/j.msec.2020.111514

Shamsipur, M., Pourmortazavi, S. M., Beigi, A. A. M., Heydari, R., & Khatibi, M. (2013). Thermal stability and decomposition kinetic studies of acyclovir and zidovudine drug compounds. AAPS PharmSciTech, 14(1), 287–293. https://doi.org/10.1208/s12249-012-9916-y

About Figure 3B and the EDS results: The EDS analysis was carried out in order to verify if the particles observed through Scanning Electron Microscopy were from acyclovir. This was another characterization method to analyze the drug incorporation and support the results. The spectrum shown in Figure 3B is also generally exposed in the literature to describe the elements identified through EDS analysis. The ethylene vinyl acetate does not present any nitrogen on its structure, so the particles containing nitrogen could be recognized as drug particles. However, the regions of EVA matrix presented a small percentage of nitrogen. In the EDS spectrum, as explained in L 205-206, the carbon and nitrogen peaks overlap, which reduces the accuracy of nitrogen quantification. Therefore, the authors considered it important to show the obtained peaks to explain why this nitrogen amount was found at EVA matrix regions.

The FTIR analysis is widely used to identify substances as it investigates molecular vibrations. The regions highlighted in the graph describe some characteristic bands of acyclovir that were detected into the EVA matrix. Moreover, it was possible to verify that the intensity of these peaks was higher for the sample with 20 wt.% of ACV compared with the sample with 10 wt.%. Therefore, this data not just confirms the presence of acyclovir but also indicates the difference between the sample’s drug concentration. For this reason, the authors considered that this section is relevant to support our results.

  1. I am concerned about the data in figure 6 it appears the vast majority of the ACV is immediately released on day 1. It does suggest tiny amounts are released later after day 1, but are these at levels of any effective use?    In addition, the release experiment is somewhat flawed as the study tried to evaluate what would be released in a “in use” setting.  But the study was done at pH 7.4.  the vaginal pH is generally considered to be a more acidic ranging from 3.5-5.   At minimum the study should be redone at this relevant pH as for amounts released over time (right figure 6 panel) maybe a log graph might be better, as nothing can be seen to be released after day 20. How does this compare with a dose of topical ACV or ACV in a liquid matrix, such as a cream?  There are no studies done to indicate an improvement over current protocols and uses.  and the lack of any animal model testing demotes the study importance

Answer: The burst-release is known as this accelerated release during the first day of application. This behavior is generally observed in matrix devices (like the ones in this work) and was reported by many authors as described through the text. Moreover, this is a characteristic behavior of EVA as described in the literature and referenced through the text (L 280-281). The main reason for this event is the drug particles located at the surface that are rapidly available to the release media. Furthermore, the materials used as a matrix, the physical state of the API (i.e. solid or melted), the matrix permeability and the API’s dispersion throughout the matrix are directly related with the intensity of this event. From Figure 6 (left), the accumulated release shows that despite the burst-release at the first 24 hours, a controlled release continues at constant rates up to 80 days. On the other hand, the daily graph (Figure 6 – right) is important to confirm the presence of the burst release. This behavior is very similar to other studies of EVA/ACV devices reported in literature (Genina et al. (2016) and Giannasca et al. (2020)). However, the present work followed the release for longer periods than the previous studies, which provide a better overview for long-term applications.

This methodology was chosen due to the properties of the solution used in the in-vitro tests with viral replication. This solution presents a pH around 7 which is quite similar to the PBS used in this work than the vaginal fluid. Furthermore, the PBS solution is widely used in drug release monitoring which may contribute to further comparisons of the release behavior, especially in terms of the EVA performance as a matrix device. In addition, the data allowed us to analyze drug release behavior and apply the results through mathematical modeling to predict many important parameters.

Finally, we understand that in-vivo testing could be important to validate, given a real perspective for the produced devices. However, as the 3D printing of drug delivery systems is still an innovation there are still some barriers to overcome before starting these kinds of analyses. Recently, Kumar Gupta et al. (2022) (reference was included into the text) presented a great review about the real advantages and challenges of using 3D printing techniques in the pharmaceutical industry. As they concluded: “[…] there are also several unanswered questions mainly in terms of regulatory perspectives, which are expected to be acknowledged with the confirmation of different categories of products and evolution of regulatory guidelines”. Most of the works found in the literature just performed the drug release monitoring. In the present work, the authors performed some biological assays which we believe is already a step forward for the additive manufacturing research area. Therefore, as one of the main objectives of this work was also to provide information and to contribute to the evolution of this disruptive manufacturing technique, the results can help to produce even more reliable devices that may be used in in-vivo tests in the future.

References:

Genina, N., Holländer, J., Jukarainen, H., Mäkilä, E., Salonen, J., & Sandler, N. (2016). Ethylene vinyl acetate (EVA) as a new drug carrier for 3D printed medical drug delivery devices. European Journal of Pharmaceutical Sciences, 90, 53–63. https://doi.org/10.1016/j.ejps.2015.11.005

Giannasca, N. J., Suon, J. S., Evans, A. C., & Margulies, B. J. (2020). Matrix-based controlled release delivery of acyclovir from poly-(ethylene co-vinyl acetate) rings. Journal of Drug Delivery Science and Technology, 55(October 2019), 101391. https://doi.org/10.1016/j.jddst.2019.101391

Kumar Gupta, D., Ali, M. H., Ali, A., Jain, P., Anwer, M. K., Iqbal, Z., & Mirza, M. A. (2022). 3D printing technology in healthcare: applications, regulatory understanding, IP repository and clinical trial status. In Journal of Drug Targeting (Vol. 30, Issue 2, pp. 131–150). Taylor and Francis Ltd. https://doi.org/10.1080/1061186X.2021.1935973

Reviewer 2 Report

The manuscript describes in detail the manufacture of two devices for the delivery of acyclovir (ACV) as a treatment for genital herpes simplex virus (HSV) infections.  The principal focus is the design, manufacture and structural and drug delivery characteristics of the devices, which are well presented and highly relevant.

The principal criticism of the data presented is the antiviral studies.  The authors outline the experimental design, which is appropriate for the studies.  However, the data are presented in a manner that is confusing and difficult to interpret.  The table (Table 5) gives a single value of virus input and virus recovery, but only in generalized Log10 levels.  There are no error bars nor any indication in the table that addresses the number of replicates within an experiment, nor how many times the experiment was performed.  These data are insufficiently presented to draw any conclusions regarding the efficacy of the drug release capabilities, as measured by antivral activity, of the devices in this in vitro culture system.  The data would be better presented in graphical format including error bars, with a detailed experimental design in the figure legend.

Author Response

Response to Reviewer 2 Comments

The manuscript describes in detail the manufacture of two devices for the delivery of acyclovir (ACV) as a treatment for genital herpes simplex virus (HSV) infections.  The principal focus is the design, manufacture and structural and drug delivery characteristics of the devices, which are well presented and highly relevant.

The principal criticism of the data presented is the antiviral studies.  The authors outline the experimental design, which is appropriate for the studies.  However, the data are presented in a manner that is confusing and difficult to interpret.  The table (Table 5) gives a single value of virus input and virus recovery, but only in generalized Log10 levels.  There are no error bars nor any indication in the table that addresses the number of replicates within an experiment, nor how many times the experiment was performed.  These data are insufficiently presented to draw any conclusions regarding the efficacy of the drug release capabilities, as measured by antivral activity, of the devices in this in vitro culture system.  The data would be better presented in graphical format including error bars, with a detailed experimental design in the figure legend.

Answer: We agree with the review. The Table 5 was replaced by Figure 7. The error bars were included, considering experiments in independent triplicates.

Reviewer 3 Report

In general, the article makes a good impression, the study was carried out logically and consistently.
I have only one comment:
The authors do not write anything about the prospect of further research into their development in clinical trials. This section should be added at the end.  

Author Response

Response to Reviewer 3 Comments

In general, the article makes a good impression, the study was carried out logically and consistently.
I have only one comment: The authors do not write anything about the prospect of further research into their development in clinical trials. This section should be added at the end. 

Answer: The suggestion was accepted and some sentences about further research were included in the conclusion section (L338-346). “It is important to notice that 3D printing of drug delivery devices is still a disruptive process and there are still many challenges, especially in terms of safety and regulation. Many authors have been exploring these techniques, providing relevant information to improve this technology. The results obtained in this work demonstrated the potential of these materials for long term genital herpes treatment. Therefore, it indicates the feasibility of the fused filament fabrication process to produce personalized and more efficient solutions for the biomedical and pharmaceutical industry. For future research, new materials can be tested to compare performance between different devices and clinical trials should be considered to validate the experiments performed”

One of the main steps of additive manufacturing is the materials selection. The type of matrix directly affects the device’s properties such as the drug release behavior and the biocompatibility. Therefore, the authors have been investigating other materials for the same application in order to compare the performance of different devices. However, as the study is under development, we are not allowed to present any detailed information. Therefore, we just mentioned the possibility to test different materials as described above.

Reviewer 4 Report

de  Carvalho Rodrigues at. al In the manuscript „Addative manufacturing of EVA….“ describe manufacturing of EVA 3D printed devices with incorporated acyclovir. The main part of the manuscript focuses on the properties of the printed material (structure and the content confirmation) and authors demonstrate successful release of the active compound and its antiviral activity.

The manuscript is offering a novel approach in treatment of recurring HSV-2 infections and has some potential as preemptive therapy. The manuscript is well written, easy to read.

 The description of cytotoxicity and antiviral assays section can be improved. It is not clear which part, size of the tested sample was used for these assays. Table 5 shows unusual PfU/ml? 5 Log ? 5 Log 10? In how many replicates these experiments were performed. Were different parts of divece used?

 Minor:

 Figure 1. lower quality; difficult o observed difference

Introduction: 1 p line 35.- the sentence needs revision. Latent viruses do not spread, rather through limited productive infection (undetected /subclinical).

Guanosine family? Family of nucleoside analogs based on guanosine.

Author Response

Response to Reviewer 4 Comments:

de  Carvalho Rodrigues at. al In the manuscript „Addative manufacturing of EVA….“ describe manufacturing of EVA 3D printed devices with incorporated acyclovir. The main part of the manuscript focuses on the properties of the printed material (structure and the content confirmation) and authors demonstrate successful release of the active compound and its antiviral activity.

The manuscript is offering a novel approach in treatment of recurring HSV-2 infections and has some potential as preemptive therapy. The manuscript is well written, easy to read.

 The description of cytotoxicity and antiviral assays section can be improved. It is not clear which part, size of the tested sample was used for these assays. Table 5 shows unusual PfU/ml? 5 Log ? 5 Log 10? In how many replicates these experiments were performed. Were different parts of divece used?

Answer: We agree on revision. Table 5 was replaced by Figure 7. The error bars were included, considering experiments in independent triplicates.

Minor:

Figure 1. lower quality; difficult o observed difference

Answer: The suggestion was accepted. The figure resolution has been improved and we removed the following sentence “Furthermore, the EVA-P20 samples were slightly whiter than EVA-P10 due to its higher drug concentration.” because we agreed that it was not possible to see this difference in this Figure, as described by the reviewer. However, the aim of this image was to show that the devices were successfully printed with both a simple geometry (Intravaginal ring) and a more complex design (IUD). As 3D printing is still an innovative technique to produce drug delivery devices, the papers about this subject generally expose the printed devices in order to show the advances in terms of quality and complexity of their products. In resume, it is just to provide some general information about the prototype. Moreover, it is possible to identify the difference between the pure sample (EVA) and drug loaded devices (EVA/ACV) as the first is transparent and the others are white and opaque.

Introduction: 1 p line 35.- the sentence needs revision. Latent viruses do not spread, rather through limited productive infection (undetected /subclinical).

Answer: The suggestion was accepted. This information was removed

Guanosine family? Family of nucleoside analogs based on guanosine.

Answer: The sentence was corrected as follows L 38-39: “Acyclovir (ACV) is a highly efficient antiviral agent against HSV. It is a guanosine analogue […]”.

Round 2

Reviewer 1 Report

The revise article is accompanied by an extensive rebuttal that explains the basis for doing many of the biophysical studies. A lot of the rebuttal makes the studies clearer to the reviewer.  However, they do not nescessarily make it clearer to the reader, bacuase some of the issues brought up are not well incorporated into the manuscript.  I would like to suggest that the authors consider incorporating many of the clarifying statements into the manuscript that I read in the rebuttal.  After reading them, I was better informed.  However the paper reader may not be so fortunate…   

English and grammar are much better

Line 178, should insert “scanning EM studies were done because…..

200 analyses

209 suggest insert some of the information regarding ACV melting temperature  Acyclovir generally starts its thermal degradation at temperatures above 400 °C, as described by Celebioglu & Uyar (2021) and Shamsipur et al. (2013) Therefore, there was suspected to be no thermal degradation of ACV during the extrusion and/or printing process. Thermal analysis was carried from -40 to 280 °C to investigate the drug incorporation. 

L229 melting point (~400oC)

lL250 maybe insert some of the rebuttal  explanation after the first line FTIR analysis is widely used to identify substances as it investigates molecular vibrations. The regions highlighted in the graph describe some characteristic bands of acyclovir that were detected into the EVA matrix. Moreover, it was possible to verify that the intensity of these peaks was higher for the sample with 20 wt.% of ACV compared with the sample with 10 wt.%.

statement on line 340 needs references

Author Response

Revisor 1:

English and grammar are much better

Answer: Thanks.

Line 178, should insert “scanning EM studies were done because…..

Answer: The information was added.  

200 analyses

Answer: Ok.

209 suggest insert some of the information regarding ACV melting temperature  Acyclovir generally starts its thermal degradation at temperatures above 400 °C, as described by Celebioglu & Uyar (2021) and Shamsipur et al. (2013) Therefore, there was suspected to be no thermal degradation of ACV during the extrusion and/or printing process. Thermal analysis was carried from -40 to 280 °C to investigate the drug incorporation. L229 melting point (~400oC)

Answer: The information was added.  

lL250 maybe insert some of the rebuttal  explanation after the first line FTIR analysis is widely used to identify substances as it investigates molecular vibrations. The regions highlighted in the graph describe some characteristic bands of acyclovir that were detected into the EVA matrix. Moreover, it was possible to verify that the intensity of these peaks was higher for the sample with 20 wt.% of ACV compared with the sample with 10 wt.%.

Answer: We appreciate the reviewer's suggestion. The manuscript describes: 
“The successful loading of ACV in EVA devices was verified through the identification of characteristic absorbance peaks of the drug by ATR-FTIR analysis. ACV shows an absorbance region between 3200 cm-1 and 3600 cm-1 mainly due to N-H primary and secondary amine stretching or to O-H stretching [14,42]. The 1738 cm-1 and 1634 cm-1 peaks are related to the C=O stretching and N-H primary amine bending, respectively [14,43]. Moreover, the 1542 cm-1 band represents C=N and C-N stretching [42]. The absorbance bands at 1186 cm-1 and 1107 cm-1 are associated with bending of amine groups and C-O stretching [42]. The 785 cm-1 peak occurs due to the out-of-plane bending of C-H groups and 685 cm-1 to the N-H wagging [12,14]. The EVA characteristic FTIR spectra shows some absorbance peaks between 2928 cm-1 and 2849 cm-1 related to the asymmetric and symmetric stretching of -CH2- groups, respectively [44]. The 1736 cm-1 peak is related to the C=O vibrations [28,44]. Furthermore, the bands at 1463 cm-1 and 1370 cm-1 are associated with deformation in the plane of -CH2- and -CH3- in the plane [45]. Finally, the peak at 718 cm-1 refers to long chains -CH2- present in polyethylene (PE) [45].”

statement on line 340 needs references

Answer: The paragraph has been reworded to be conclusive and not lead to discussion.

“The results obtained in this work demonstrated the potential of these printed devices for long-term genital herpes treatment. Therefore, it indicates the feasibility of the fused filament fabrication process to produce personalized and more efficient solutions for the biomedical and pharmaceutical industries. For future research, new materials can be tested to compare performance between different devices and clinical trials should be considered to validate the experiments performed.”

Reviewer 2 Report

The authors have done an excellent job of addressing the concerns of previous reviews.  The new title is an improvement and better describes the scope of the research.  The presentation of the viral titers in graphic form, with the inclusion of error bars and the information in the Materials and Methods related to experimental replicates has strengthened the validity of the study.

Author Response

Revisor 2:

The authors have done an excellent job of addressing the concerns of previous reviews.  The new title is an improvement and better describes the scope of the research.  The presentation of the viral titers in graphic form, with the inclusion of error bars and the information in the Materials and Methods related to experimental replicates has strengthened the validity of the study.

Answer: Thank you for your positive feedback